# Anomalous magnetoresistance by breaking ice rule in Bi$_2$Ir$_2$O$_7$/Dy$_2$Ti$_2$O$_7$ heterostructure

Han Zhang[1,7], Chengkun Xing[1,7], Kyle Noordhoek[1,7], Zhaoyu Liu[2], Tianhao Zhao[3], Lukas Horák[4], Qing Huang[1], Lin Hao[1], Junyi Yang[1], Shashi Pandey[1], Elbio Dagotto[1,5], Zhigang Jiang[3], Jiun-Haw Chu[2], Yan Xin[6], Eun Sang Choi[6], Haidong Zhou[1] ✉ & Jian Liu[1] ✉

While geometrically frustrated quantum magnets host rich exotic spin states with potentials for revolutionary quantum technologies, most of them are necessarily good insulators which are difficult to be integrated with modern electrical circuit. The grand challenge is to electrically detect the emergent fluctuations and excitations by introducing charge carriers that interact with the localized spins without destroying their collective spin states. Here, we show that, by designing a Bi$_2$Ir$_2$O$_7$/Dy$_2$Ti$_2$O$_7$ heterostructure, the breaking of the spin-ice rule in insulating Dy$_2$Ti$_2$O$_7$ leads to a charge response in the conducting Bi$_2$Ir$_2$O$_7$ measured as anomalous magnetoresistance during the field-induced Kagome ice-to-saturated ice transition. The magnetoresistive anomaly also captures the characteristic angular and temperature dependence of this ice-rule-breaking transition, which has been understood as magnetic monopole condensation. These results demonstrate a novel heteroepitaxial approach for electronically probing the transition between exotic insulating spin states, laying out a blueprint for the metallization of frustrated quantum magnets.

Novel emergent states of quantum magnets often arise from geometric frustration, where local spins residing on the corners of a triangle or tetrahedron cannot agree on any alignment that simultaneously minimizes their energies[1]. Such a spin lattice has a large degree of degeneracy due to an enormous number of configurations at the same or similar energy, leading to exotic ground states, entangled fluctuations, and collective excitations[2–7], such as magnetic monopole and spinon, which may afford revolutionary quantum technologies[8–10]. The so-called dipolar spin ice is one of the most iconic representatives of such frustrated magnets. It was first discovered on pyrochlore lattices of Ising spins constrained along the local [111] direction, such as Ho$_2$Ti$_2$O$_7$ and Dy$_2$Ti$_2$O$_7$[11–13]. While the spin-spin correlation is dominated by the dipolar interaction, the ferromagnetic instability is geometrically frustrated by the local Ising anisotropy[1]. As a result, the

ground state of each tetrahedron settles in one of the six-fold degenerate 2-in-2-out configurations, forming a spin ice network throughout the lattice following the ice rule in analog with the proton displacement in the water ice[14]. The non-zero residual entropy has indeed been observed with a value close to that in the water ice[15]. Breaking the ice rule could lead to rich spin dynamics, including emergent magnetic monopoles[16]. For instance, applying a magnetic field along the pyrochlore [111] direction turns the three-dimensional spin ice into a Kagome spin ice[17], and eventually stabilizes the 3-in-1-out/1-in-3-out configuration by condensing the monopoles[1,2,18], giving rise to a jump from ~3.3 $\mu_B$/Dy to ~5 $\mu_B$/Dy in Dy$_2$Ti$_2$O$_7$ between two magnetization plateaus[19]. The 3-in-1-out/1-in-3-out state is effectively a magnetic charge-ordered state, which is also called the saturated ice[20] where the ice rule is broken everywhere.

[1]Department of Physics and Astronomy, University of Tennessee, Knoxville, TN 37996, USA. [2]Department of Physics, University of Washington, Seattle, WA 98195, USA. [3]School of Physics, Georgia Institute of Technology, Atlanta, GA 30332, USA. [4]Charles University, Prague 11636, Czechia. [5]Materials Science and Technology Division, Oak Ridge National Laboratory, Oak Ridge, TN 37831, USA. [6]National High Magnetic Field Laboratory, Florida State University, Tallahassee, FL 32310, USA. [7]These authors contributed equally: Han Zhang, Chengkun Xing, Kyle Noordhoek. ✉e-mail: hzhou10@utk.edu; jianliu@utk.edu

However, the spin-ice behaviors have been only observed in highly insulating compounds[1,2,11], where the spins are hosted by highly localized electrons in the absence of itinerant carriers, such as the $f$-electrons of $Ho^{3+}$ or $Dy^{3+}$ ions in combination with the empty $d$-shell of the $Ti^{4+}$ valence state in the pyrochlore titanate[11–13]. When introducing itinerant carriers, simply replacing the B-site with another element with a metallic configuration often ends up interfering the spin ice state. For instance, the correlated $d$-electrons in pyrochlore iridates not only exhibit an all-in-all-out ordering of their own at much higher temperatures, but also force the same magnetic order onto the A-site sublattice regardless of the rare earth element[21]. In some cases, the interactions within the rare earth sublattice may modify the transport properties of the $d$-electron[22–24] at lower temperatures, such as the fragmented monopole crystal ground state[25] as well as the monopole-density-related magneto-transportation[26]. However, the spin state of the rare earth sublattice is often impaired by the energy scale of the $d$-electrons, which is much larger than the interaction. This situation indeed applies to a broad class of frustrated magnets beyond spin ice.

In this work, we report the observation of an anomalous magnetoresistance (MR) in epitaxial pyrochlore heterostructures of $Bi_2Ir_2O_7$/$Dy_2Ti_2O_7$ (BIO/DTO), where DTO hosts the spin ice state and BIO, the only nonmagnetic/time-reversal-invariant member of the pyrochlore iridate series, provides the correlated carriers. We directly grow ultrathin BIO films on DTO single crystals as illustrated in Fig. 1a, b. A cusp-like MR anomaly is observed, when the field induces the transition between the Kagome spin ice state and the saturated ice. These results demonstrate an epitaxial interface approach for enabling charge responses and detection of the exotic spin state transitions in insulating frustrated magnets.

## Results

DTO single crystal substrates were prepared by the floating zone method[27]. The crystals were first oriented and cut into substrate pieces along the DTO (111) lattice plane. The substrate surfaces were then polished through a five-step process (see details in experimental method) to achieve atomically flat surfaces. Figure 1c shows a typical atomic force microscope (AFM) image of the DTO single crystal substrate surface after the process, with a root mean square surface roughness of ~1.21 Å. Thin BIO layers were deposited on the DTO substrates by pulsed laser deposition[28]. The BIO/DTO interface was examined with the scanning transmission electron microscopy (STEM). A typical cross-section image in Fig. 1d shows a sharp epitaxial

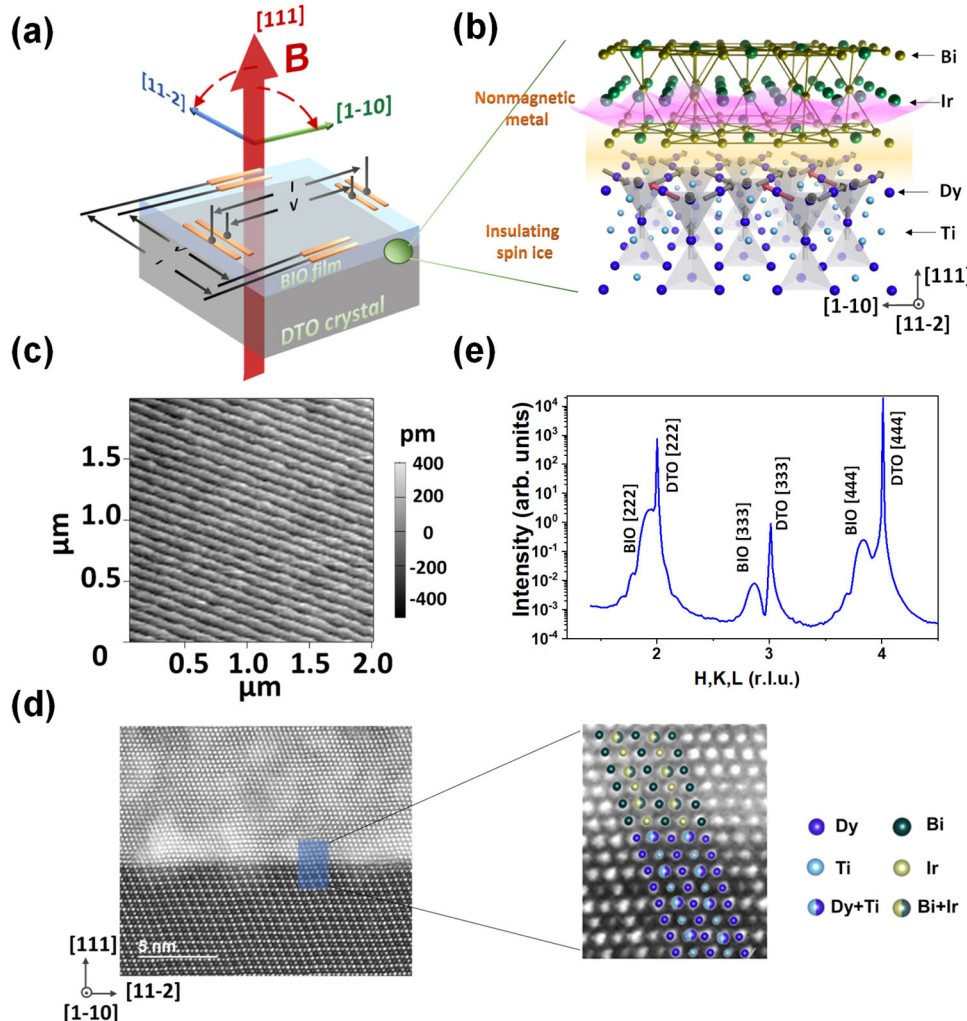

**Fig. 1 | $Bi_2Ir_2O_7$/$Dy_2Ti_2O_7$ heterostructure. a** Schematic diagram of the $Dy_2Ti_2O_7$/$Bi_2Ir_2O_7$ heterostructure and the measurement configuration. **b** A schematic drawing showing a zoomed region across the interface. The local spins (gray) are shown on the DTO sides with a few of them (red) flipped to break the ice rule and form the '3-in-1-out/1-in-3-out' configuration. A wavy surface curve (pink) is shown on the BIO side to denote charge carrier. **c** A typical AFM image of DTO substrate surface with atomically flat terraces. **d** Transmission electron microscopy image with [1-10] being the lamella surface normal shows the sharp BIO/DTO interface of the heterostructure. The white patches are from focus iron beam milling (see Materials and Methods). A thicker film sample with thickness ~10.5 nm was used in the TEM study to protect the interface. **e** An X-ray diffraction scan covering the DTO [222] to [444] peaks of a BIO/DTO heterostructure.

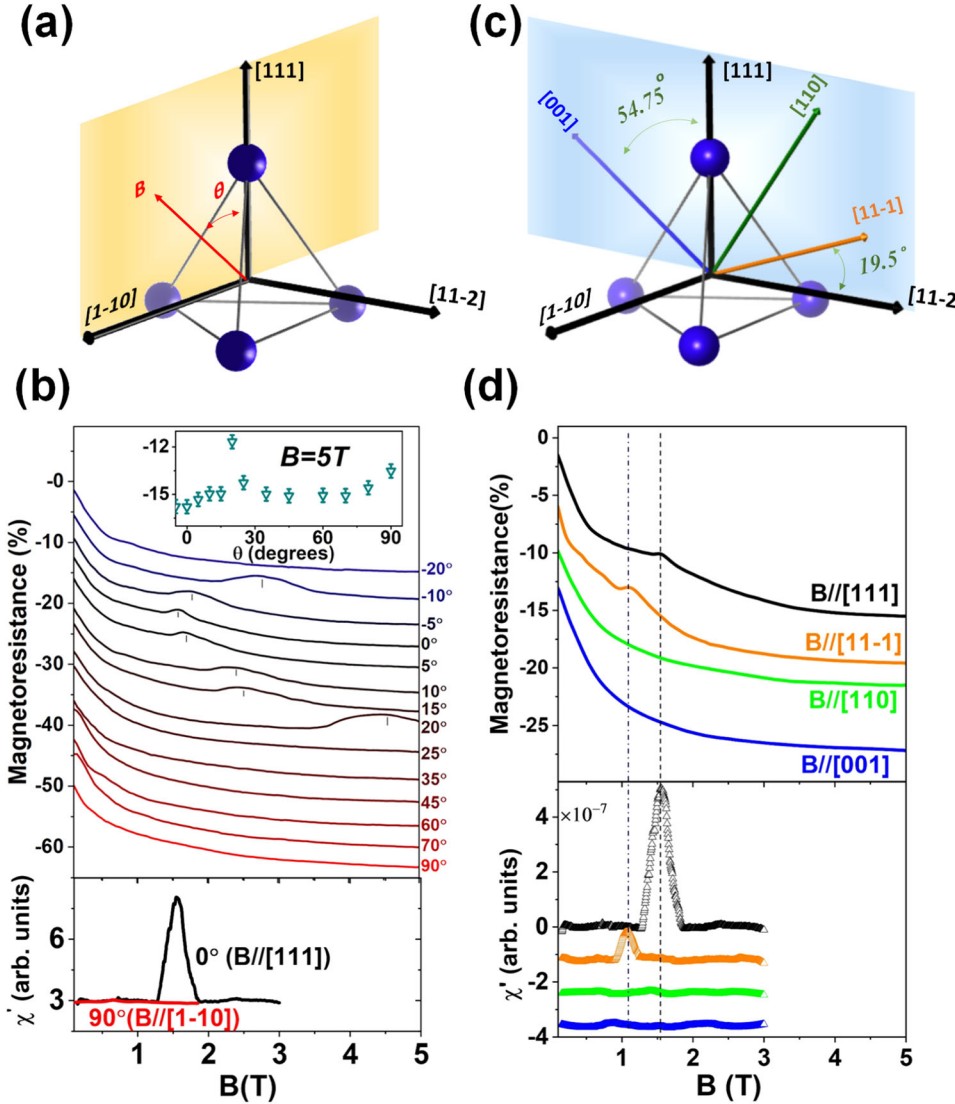

**Fig. 2 | Magnetoresistance anomaly. a** Geometric diagram of the MR measurement with **B** in the (11-2) plane, where $\theta$ is the field angle respected to the [111] direction. MR is defined as MR = $\frac{R(B)-R_0}{R_0} \times 100\%$, where $R_0$ is the zero-field resistance. **b** MR (top) and AC susceptibility (bottom) at 0.03 K at selected angles; inset illustrates the MR at each angle when **B** = 5 T. **c** Geometric diagram of the MR measurement with **B** in the (1-10) plane, which contains four high-symmetry axes. **d** MR (top) and AC susceptibility (bottom) at 0.03 K along the high-symmetry axes in this geometry. Vertical offsets are applied on the curves for clarity.

interface. Energy dispersive X-ray spectroscopic (EDS) map (Supplementary Fig. 1) further confirms no significant interdiffusion across the interface. Figure 1e shows a typical specular X-ray diffraction scan covering the DTO [222] to [444] structural peaks, which are accompanied with the corresponding peaks of a BIO film of 3.4 nm, confirming the epitaxial growth. Representative reciprocal space mapping (RSM) shows that the BIO film is in a fully strained state (Supplementary Fig. 2a). The film thickness is extracted from fitting the X-ray reflectivity curve (Supplementary Fig. 2b, c), which also yields a film density consistent with expectation.

While the all-in-all-out magnetic ordering of the Ir pseudospin-1/2 electrons in the pyrochlore iridates has an increasing ordering temperature with decreasing A-site ionic radius[29], BIO is a bad metal on the paramagnetic side of the quantum critical point[30]. No magnetic ordering or spontaneous time-reversal-symmetry-breaking was found down to 50 mK[31]. Moreover, it has been shown that, although the BIO metallicity will degrade when grown as ultrathin films likely due to weak localization and/or disorder-enhanced electron-electron interaction in two-dimensions, the MR remains rather isotropic when the field is applied along different

crystallographic axes[32]. Our ultrathin BIO films on DTO indeed show a weakly increasing resistivity with lowering temperature as well as an isotropic MR (Supplementary Fig. 2d, e), consistent with previous reports[32]. No temperature-induced transition is observed, indicating that the BIO films remain nonmagnetic. However, an emergent anisotropic anomaly was observed while measuring MR below 1 K. Figure 2a, b shows MR measurements at 0.03 K with the magnetic field **B** along different directions in the (11-2) plane, while the current is always along the [11-2] axis perpendicular to the field (Fig. 1a). One can see the overall MR response with **B**//[111] is similar to **B**//[1-10] except a positive cusp-like anomaly around 1.5 T. This anomaly was confirmed in multiple samples (Supplementary Table 2 and Supplementary Fig. 2f). It can also be seen that the anomaly has a reduced size in samples with a much thicker BIO film, indicating that the anomaly is caused by the interface with DTO (Supplementary Table 2 and Supplementary Fig. 3a). Our AC susceptibility measurement, where the signal is dominated by the bulk of the DTO substrate, shows that this field corresponds to the critical field **B**$_c$ of the Kagome spin ice-to-saturated ice transition of DTO, and it is higher than the intrinsic value of **B**$_c$ ~ 1 T[20] due to the

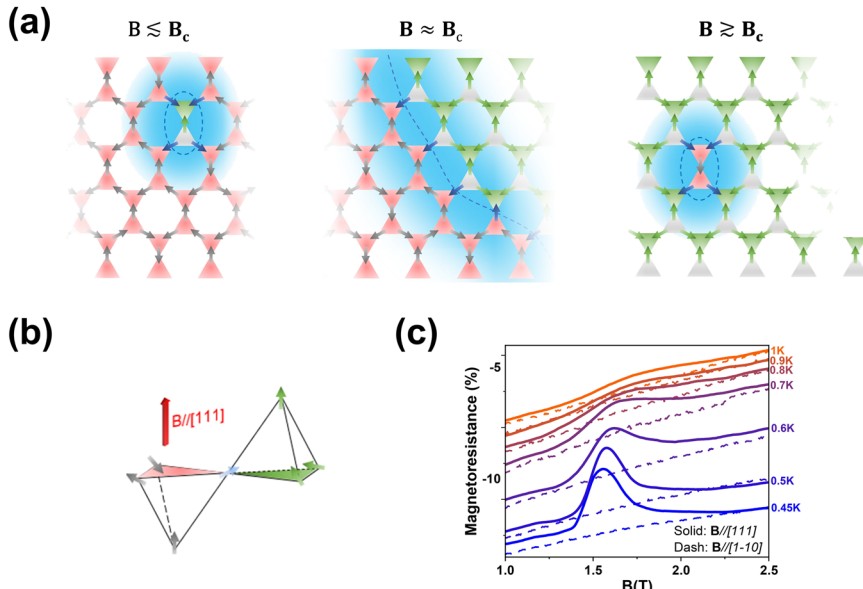

**Fig. 3 | Phenomenological mechanism. a** Schematics showing the boundary in the Kagome layer between the Kagome spin ice and the saturated ice state. The tetrahedra with the 2-in-2-out configuration and the 3-in-1-out (1-in-3-out) configuration are in red and green, respectively. The blue arrows denote the sites that connect the two regions with the area highlighted in light blue indicating the boundary region where the spins are less stable. A three-dimensional schematic is showed in panel **b** for a pair of the tetrahedra across the boundary. **c** MR in the region of the anomaly from 0.45 to 1 K. The solid and dashed lines were measured with **B**//[111] and **B**//[1-10], respectively. The curves are vertically shifted for clarity and comparison purposes.

demagnetization effect of the plate-shape of the crystal substrate (see Fig. 1a and Supplementary Fig. 4).

Moreover, as we rotate **B** away from [111], the anomaly displays a clear shift to higher fields with the width of the feature being broadened as well, which is consistent with the fact that the field projection to [111] is decreasing as the field angle increases. A larger external field is thus needed to induce the transition in DTO, and the finite width of the transition is effectively stretched. Moreover, the observed increase is rapid and clearly does not follow any simple cosine function, which is a signature of the field-induced transition in spin ice due to geometric frustration[33,34]. Deriving $B_c$ from the dipolar spin ice model indeed shows that $B_c$ diverges when the field deviates from [111] by ~20° (Supplementary Fig. 5), which is consistent with the fact that the anomaly is quickly shifted beyond the measured field range and the shape of the MR curve by that point is virtually the same as that with **B**//[1-10]. (The asymmetric behavior between the positive and negative angles is due to a small instrumental misalignment ~2°, see Supplementary Fig. 5 for details). The angle-controlled transition is further illustrated in the inset of Fig. 2b, where the MR at 5 T is extracted and plotted against the field angle from [111] ($\theta = 0°$) to [1–10] ($\theta = 90°$). One can see a sharp increase of the resistivity around the critical angle $\theta_c$ (~20°) due to the anomaly. Interestingly, the resistivity on both sides of $\theta_c$ shows a much smaller but visible difference, which can be attributed to the fact that DTO remains a spin ice at 5 T with $\theta > \theta_c$ while becoming a saturated ice with $\theta < \theta_c$. The remnant spin disorder in the former is likely to elevate the resistivity compared with the latter.

Figure 2c, d shows representative MR curves of the BIO/DTO heterostructure at 0.03 K measured in another configuration where **B** is applied in the (1-10) plane with current along the [1-10] axis (Fig. 1a). This setup allows the field to access multiple high-symmetry axes, including [110], [11-1], and [001], in addition to [111] as shown in Fig. 2c. One can see that, while the overall MR response remains similar for all directions, the MR curve with **B**//[11-1] clearly shows an extra positive anomaly that is absent with **B**//[110] and **B**//[001] but is similar to the case of **B**//[111]. This observation is consistent with the fact that [111] and [11-1] are two equivalent axes in DTO. The difference is that the anomaly with **B**//[11-1] is at ~1 T, which is lower than that with **B**//[111]

and close to the intrinsic $B_c$. We again confirmed this behavior by AC susceptibility measurements, which show in the bottom panel of Fig. 2d that the peaks in correspondence to the transition for **B**//[111] and **B**//[11-1] indeed occur at different fields and match the fields of their MR anomalies, respectively. The shift of $B_c$ between [111] and [11-1] is attributed to the much stronger demagnetization effect for the surface normal direction [111]. By comparison, the AC susceptibility is featureless for **B**//[110] and **B**//[001]. All the observations above corroborate that the MR anomaly of the BIO layer is indeed a result of the field-induced ice-rule-breaking transition in the insulating DTO substrate and is remarkably sensitive to any shift of $B_c$. Note that the observed positive cusp-like anomaly cannot be attributed to the DTO stray field on BIO because the magnetization jump of the DTO transition would increase the effective field on BIO and lead to a negative step-like MR jump instead. In addition, the resistivity at 5 T with **B**//[110] is also larger than that of **B**//[111] and [11-1] (Supplementary Fig. 2g), consistent with the impact of the remnant spin disorder in the spin ice state as discussed with the previous configuration.

## Discussion

This result demonstrates that forming a heterostructure can effectively introduce interactions between the charge carriers from one side and the frustrated spins on the other side. The excellent consistency in the critical field between MR and the bulk-sensitive AC susceptibility reveals the robustness of this method. Note that the field, regardless of its direction, always locally flips the moments to partially lift the 2-in-2-out degeneracy of the Dy tetrahedron within the spin ice state. Yet the MR anomaly only occurs when the transition to the saturated ice state is induced, indicating that the charge carriers are particularly sensitive to the ice-rule-breaking spin flips. Given that the Kagome spin ice-to-saturated ice transition is a first order transition[1,19,20], a phenomenological mechanism of the anomalous MR responses can be depicted with the coexistence of the two spin states in the Kagome plane perpendicular to the field (since the triangular plane is polarized in both cases). As illustrated in Fig. 3a, a boundary of the two regions is defined by the spins that connect tetrahedra hosting the 2-in-2-out and 3-in-1-out/1-in-3-out configuration (Fig. 3b), respectively. The spins nearby

the boundary are most unstable and fluctuating, thus potentially introducing extra scattering to the charge carriers when entering one region from another which increases the resistivity. Figure 3a shows a cartoon of how the density of these boundaries may increase and decrease across the critical regime, capturing the enhanced resistivity during the transition manifested by the positive sign and the cusp shape of the anomaly. As discussed earlier in the angle-dependent data, although the different degrees of spin disorder of the two states may still induce a difference in the resistivity outside the transition region, this effect is smaller than the anomaly likely due to the fact that local moments are more stable away from the transition.

To further verify this picture, we measured a 3.7 nm BIO film on nonmagnetic $Y_2Ti_2O_7$ substrate as a reference sample and observed isotropic MR as functions of temperature and magnetic field without any anomaly (Supplementary Fig.6). This comparison confirms that DTO influences the transport properties of the BIO film mainly in its transition regime. In other words, the anomaly is a charge response to the magnetic fluctuations in the transition region, and it has a peak-like line shape similar to the AC susceptibility. Although a microscopic theory is yet to be developed to describe the interaction of the Dy moment and the BIO carriers, it likely depends on the hybridization between the localized *f*-states at the Dy sites and the itinerant states of Ir *d*-electrons since such an interaction is usually captured by the Anderson Model[35,36] for metals with local moments. The difference here is that the localized moments and the itinerant carriers are on opposite sides of the heterostructure and they couple across the interface. We note that the spin Hall effect (SHE) in metals may also produce anisotropic MR responses through the so-called spin hall magnetoresistance when in direct contact with an insulating magnet[37]. However, SHE does not require the field-induced transition and does not account for the fact that the anomaly only occurs near the critical field of the transition, excluding SHE and any other process irrelevant to the field-induced transition from being the underlying mechanism. In fact, spin hall magnetoresistance requires an orthonormal geometry among the surface normal, current direction and magnetization, which is obviously unnecessary for the observed anomaly in the BIO/DTO heterostructure. In addition, the observed anomaly could not be attributed to the magnetocaloric effect as it is expected to cause responses of opposite signs when sweeping the magnetic field upward and downward, which was not observed in our experiment.

Studies of the field-temperature phase diagram of spin ice have shown that the field-induced phase boundary is terminated at elevated temperature and the transition becomes a continuous crossover as there is no sharp distinction between the Kagome spin ice and the saturated ice anymore[2,19]. This behavior has been understood with an analogy to the liquid–gas transition in the context of a transition between a low-density phase and a higher-density phase of magnetic monopoles[16–19]. For comparison, we studied the thermal evolution of the MR anomaly with **B**//[111] between 0.45 K and 1 K. The obtained result displayed as the solid lines in Fig. 3c shows that, while the anomaly overall remains around 1.5 T, it weakens in magnitude, broadens in width, and upshifts slightly in field with increasing temperature, likely due to the thermal broadening of the Kagome spin ice-to-saturated ice transition near its critical end point[1]. The dashed lines were measured with **B**//[1-10] and serve as a reference for comparison at each temperature. The anomaly eventually starts to vanish at ~0.9 K and becomes hardly visible. These observations are consistent with the field-temperature phase diagram as well as the disappearing of the spin ice state in DTO due to thermal fluctuation[14–16] around 1 K, confirming the sensitivity of the charge carriers to the abrupt change at the boundary between the two spin states until they are no longer distinguishable above the critical endpoint of the transition[19].

To summarize, the experimental observation of an emergent MR anomaly in BIO that responds to and closely tracks the Kagome ice-to-saturated ice transition in insulating DTO demonstrated that creating epitaxial heterostructures is a highly promising approach to introduce interactions between localized frustrated spins and correlated charge carriers. It opens the possibilities of electronically probing the spin ice states under other control parameters, such as pressure and epitaxial strain[38,39], as well as investigating charge conversion of the monopole condensation[16] excitations under non-equilibrium stimuli[40,41]. Recently, the magnetic monopole dynamics was detected by a magnetization noise spectrometer in the spin ice state of DTO[42–45] as well as during the transition between two 2-in-2-out states of artificial spin ice[46]. A two-dimensional monopole gas was also proposed at the heteroepitaxial interface of the 2-in-2-out and the all-in-all-out state[47]. Such behaviors may manifest in electronic transport, too, when interfaced with charge carriers, providing another detection method and a potential route to application. The interfacial approach can readily be extended to other non-conducting frustrated quantum magnets, such as spin liquid and quantum spin ice[1,3–10] for electronically detecting these exotic spin states and triggering electronic responses. The successful creation of functional heterostructure between DTO and BIO also showcases the potential of complex oxide heterostructures of the pyrochlore structure[48].

## Methods

### Synthesis of the DTO single crystal
polycrystalline DTO was prepared by thoroughly mixing stoichiometric $Dy_2O_3$ and $TiO_2$ with the molar ratio of 1:2[27]. Prior to weighing, $Dy_2O_3$ was preheated at 1000 °C for 10 h in order to remove the moisture. After grinding the mixture, sintering the mixture at 1000 °C and 1400 °C for 20 h each in air with intermediate grinding. The powders were then compressed to a feed rod with another additional annealing at 1400 °C for 20 h, which were used to grow the single crystal DTO by using a double elliptical mirror optical floating-zone furnace SC1-MDH from Canon Machinery. To avoid oxygen deficiency, we grew the single crystal in 5 atm flowing $O_2$ atmosphere. In the first cycle we used a high growth rate of 50 mm/h for both feed and seed rods and in the second cycle we used a relatively slow growth rate of 4 mm/h.

### Sample preparation
The DTO single crystal was first aligned by a HUBER Laue X-ray diffractometer and then cut into substrate pieces with DTO (111) lattice plane by using the Model 650 Low Speed Diamond Wheel Saw from South Bay Technology. The substrate surfaces were prepared by going through a five-step mechanical polishing process with monocrystalline suspension. The surfaces were characterized by an MFP-3D Atomic Force Microscope from Asylum Research. Thin film samples were synthesized by Pulsed Laser Deposition[28]. A KrF excimer laser ($\lambda = 248$ nm) was used at a repetition rate of 2 Hz and the laser beam fluency is 1.7 J/cm$^2$. The BIO films were deposited at 620 °C and 0.0667 mbar of oxygen pressure. The thickness was controlled by the nominal growth rate of BIO films at ~40 pulses/nm and verified by ex situ x-ray reflectivity. After deposition, the films were cooled down in 0.0667 mbar oxygen pressure. We focused on samples with the film thickness in the range of 3–5 nm for transport measurements.

### X-ray characterization
In-house X-ray characterization was done by an Empyrean diffractometer at Copper radiation. Synchrotron X-ray diffraction was performed at beamline 33-BM at the Advanced Photon Source at the Argonne National Laboratory.

### Transmission electron microscopes (TEM)
TEM data were acquired using a cold field emission probe-aberration-corrected JEOL JEM-ARM200cF at 200 kV with a spatial resolution of 0.078 nm. It is equipped with Oxford Aztec SDD EDS detector for elemental analysis. TEM lamella was made by focused ion beam (FIB) in

a ThermoFisher Scientific Helios G4 UC DualBeam scanning electron microscope (SEM). To avoid charging in the SEM, the thin film surface was first sputtered with a 10 nm thick layer of gold. Cross-sectional view along [1-10] was obtained by high-angle annular dark-field scanning transmission electron microscopy (HAADF-STEM) imaging. The HAADF-STEM images were collected with the JEOL HAADF detector using the following experimental conditions: probe size 7c, CL aperture 30 μm, scan speed 32 μs/pixel, and camera length 8 cm, which corresponds to a probe convergence semi-angle of 21 mrad and inner collection semi-angle of 74 mrad. EDS maps were collected in the STEM mode with a probe size of 0.12 nm.

**Resistivity and magnetoresistance measurements**

Transport measurement was performed with a Physical Properties Measurement System (PPMS) from Quantum Design (QD). The current is always applied perpendicular to the field direction. MR between 0.45 and 1 K was measured with the He-3 option of a PPMS Dynacool from QD. MR at 0.03 K was measured at the National High Magnetic Field Laboratory (NHMFL), using a 0.2 T/min sweeping rate; the data were collected at field values of a 0.05 T step size, after waiting for more than 60 s at each setpoint. All MR data were measured by utilizing the lock-in technique; The typical current applied was at the scale of 10 nA, with a frequency <20 Hz. Samples used in the measurement are of 3 mm (W) × 6 mm (L) in dimension. The DTO single crystal is of 0.5 mm in thickness.

**AC susceptibility**

AC susceptibility was measured at NHMFL. The AC susceptibility measurements were conducted with a voltage-controlled current source (Stanford Research, CS580) and lock-in amplifier (Stanford Research, SR830). The phase of the lock-in amplifier is set to measure the first harmonic signal. The rms amplitude of the AC excitation field is set to be 0.35 Oe with the frequency fixed to be 470 Hz. The measurement at NHMFL mainly focused on identifying magnetic transitions rather than obtaining absolute magnetic susceptibility values. The unit of the AC susceptibility data was given in arbitrary units because the AC susceptometer was not calibrated within the scope of the experiment.

## Data availability

The data generated or analyzed in this study are available from the corresponding authors upon reasonable request.

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

## Acknowledgements

This research is supported by the U. S. Department of Energy under grant No. DE-SC0020254. H.D.Z. and J.L. acknowledge support from the Organized Research Unit Program; the Electromagnetic Property Laboratory; the Scholarly Activity and Research Incentive Fund (SARIF) at the University of Tennessee. E.D. was supported by the US Department of Energy (DOE), Office of Science, Basic Energy Sciences (BES), Materials Sciences and Engineering Division. The work at University of Washington is supported by the David and Lucile Packard Foundation (Transport measurements) and the Air Force Office of Scientific Research through DURIP Award FA9550-20-1-0310 (Helium 3 cryostat). The transport measurement at Georgia Tech is supported by the DOE under grant No. DE-FG02-07ER46451. The synchrotron X-ray diffraction measurement used resources of the Advanced Photon Source, a U.S. Department of Energy (DOE) Office of Science User Facility operated for the DOE Office of Science by Argonne National Laboratory under Contract No. DE-AC02-06CH11357. A portion of this work was performed at the National High Magnetic Field Laboratory, which is supported by the National Science Foundation Cooperative Agreement No. DMR-1644779 and the state of Florida. The authors thank Martin Mourigal and his research group for providing help in resistivity measurements; Jenia Karapetrova for helping with the synchrotron X-ray diffraction; and Cristian Batista for fruitful discussions.

## Author contributions

H. Zhou and J.L. conceived the idea and directed the study. K.N. and C.X. performed the pulsed laser deposition. C.X. synthesized the single crystal substrates. K.N. and C.X. performed the surface preparation. L.H. and C.X. performed in-house X-ray diffraction measurement. H. Zhang performed synchrotron X-ray diffraction with the help of L.H. and J.Y., H. Zhang, Z.L., J.H.C., T.Z., Z.J., C.X., K.N., S.P., and E.S.C. performed the transport measurements. C.X. and Q.H. conducted the ac susceptibility measurement. Y.X. performed the FIB and TEM characterization. H. Zhang, K.N., C.X., H. Zhou, and J.L. analyzed the data. H. Zhang, H. Zhou, E.D., and J.L. wrote the manuscript with contributions from all other authors.

## Competing interests

The authors declare no competing interests.
