## [Peer Review File · Nature Communications]

REVIEWER COMMENTS

Reviewer #1 (Remarks to the Author):

This paper presents an experimental study on the magnetoresistance (MR) of the Bi₂Ir₂O₇ (BIO) / Dy₂Ti₂O₇ (DTO) interface. The DTO is a canonical spin-ice materials, while the BIO is a metallic magnet which also has the same pyrochlore structure. Unlike the spin-ice magnets Dy₂Ti₂O₇ or Ho₂Ti₂O₇, the itinerant electrons favor an all-in-all-out magnetic order in BIO. However, as demonstrated in this manuscript, the heterostructure formed from BIO and DTO provides a metallic thin layer whose electrical transport depend on the proximate spin-ice state in DTO, and hence can be controlled by the magnetic field.

In particular, the authors clearly showed an anomalous MR induced by a transition from the highly degenerate kagome ice phase to the fully polarized state in DTO. The experimental results and analysis seem solid to me. The reported MR anomaly is quite interesting, this work could potentially open a new avenue for meta-materials and quasi-2D systems based on spin-ice magnets.

I have a few questions/comments:

1. In Figure 2(b) and (d), the background MR (%) generally decreases with increasing magnetic field B at the temperature $T = 0.03\text{K}$. On the other hand, the MR (except for the anomaly at $B \sim 1.5\text{ T}$) in Figure 3(c) increases with increasing B -field in the temperature range $T = 0.45$ to 1K . Can the authors elaborate on why the trends of the background MR seem to be opposite at extremely low temperature ($T \sim 0.03\text{K}$) and close to the liquid-gas critical point. (Yet, interestingly, the anomaly persists in both regimes.)

2. Related to the above point is that: In the low-field kagome ice phase, the geometrical frustration results in a classical spin liquid due to the huge quasi-degenerate microscopic states governed by the kagome-ice rule. However, this is essentially a spin-disordered state. Generally, one would expect this additional disorder (either static or dynamical, due to fluctuations between different kagome ice configurations) would give rise to a larger resistivity, compared with that of the fully polarized state at high field?

Due to the rather distinct nature of the kagome-ice phase and the fully polarized state, one would expect the MR to be rather different in the two phases. However, as the authors noted, the background MR seems to follow a smooth curve (also different at low- T and T_c). Can the authors

also elaborate on this point, namely the absence of two MR-behaviors in the low and high-field phases?

I guess one possibility is that the background MR comes almost entirely from the BIO layer, so it is insensitive to the spin-state on the DTO side or the interface. And the anomalous MR at $B \sim 1.5$ T is an additional signal due to the interface?

Finally, a minor comment on the terminology. The transition of DTO at $B \sim 1.5$ T is essentially a transition from 2-in-2-out kagome ice plateau to the fully polarized state. Indeed, this transition can be viewed as a condensation of monopoles (a statement which is less trivial at $T \sim T_c$). But otherwise, this is a very simple magnetic transition, which can be easily demonstrated with one or two figures. I am not sure whether throwing terms like “monopole condensation” can help the general readers understand the underlying physics.

I would recommend the publication of this paper if the authors can address the above comments/questions.

Reviewer #2 (Remarks to the Author):

This paper presents results from an impressive materials fabrication achievement: growing a conducting pyrochlore epitaxially on top of a spin ice material. The authors demonstrate that the magnetic physics of the spin ice affects the transport properties of the conducting layer in that there is an anomaly in the magnetoresistance that corresponds to the metamagnetic transition in the spin ice. This is interesting and impressive as a materials growth advance, but I find the paper missing two critical points that would be required for publication:

1. The authors refer to a 3.4 nm and a 3.7 nm sample, but they never list all the samples that they measure or show data from the full list to allow for a sense of reproducibility in the results. This is critical, since the authors do demonstrate that the result disappears for large thicknesses as might have been expected, and some comparison would illuminate possible mechanisms. The SI should include such a list and side-by-side comparisons of the measured properties. This comparison should include some discussion of the quantitative aspects of the anomaly, as well as the qualitative similarities and differences.

2. The authors assert that the field from the spin ice cannot account for the MR anomaly because the sign is incorrect. I am not sure that is the case, given that the local fields may have interesting directional dependence, and I would have appreciated a bit more detailed explanation on this point. More importantly, however, the authors give no suggestions for what mechanisms could be coupling the two materials. Is there some correlated electron effect in the conductor that could be coming into play that somehow couples with the spin ice moments (hard to imagine)? Is the conduction happening through the spin ice in some electronic proximity effect (also hard to imagine)? Is there a strain effect associated with the transition in the spin ice (possible, but I would expect a step rather than a peak in the MR)? I can speculate as to what might be occurring here, but the authors need to include some discussion at a bare minimum for publication in any respectable journal in order to make a complete story. For publication in Nature Communications, I would expect that they would propose a mechanism and have some physics explanation for why it would produce a signal of the sort that they observe.

Reviewer #3 (Remarks to the Author):

This paper reports experiments on a heterostructure made up of two pyrochlores, the spin-ice insulator $\text{Dy}_2\text{Ti}_2\text{O}_7$, and the paramagnetic metal $\text{Bi}_2\text{Ir}_2\text{O}_7$. The authors make an excellent case that the first-order kagome-ice-to-saturated-ice transition in $\text{Dy}_2\text{Ir}_2\text{O}_7$ leads to an anomaly in the transport of the $\text{Bi}_2\text{Ir}_2\text{O}_7$. This observation highlights the very interesting possibility of devices in which the electrical detection of transitions between exotic spin states in frustrated insulators is possible. The paper can be published provided that the authors address the following comments.

1) In the introduction the authors state that

"However, the spin-ice behaviors have been only observed in highly insulating compound, where the spins are hosted by highly localized electrons in the absence of itinerant carriers, such as the f-electrons of Ho^{3+} or Dy^{3+} ions in combination with the empty dshell of the Ti^{4+} valence state in the pyrochlore titanate. When introducing itinerant carriers, simply replacing the B-site with another element with a metallic configuration often ends up destroying the spin ice state. For instance, the correlated d-electrons in pyrochlore iridates not only exhibit an all-in-all-out ordering of their own at much higher temperatures, but also force

the same magnetic order onto the A-site sublattice regardless of the rare earth element."

This is not correct. For example, $\text{Ho}_2\text{Ir}_2\text{O}_7$ and $\text{Dy}_2\text{Ir}_2\text{O}_7$ both show spin-ice behaviour on the rare-earth sublattice at the same time as all-in-all-out order on the Ir sublattice. Indeed, interactions

between the two sublattices have been shown to lead to interesting effects, including (i) a fragmented monopole crystal ground state at low temperatures [A]; and (ii) at slightly higher temperatures, a response to changes in the monopole density that can be detected electrically via the itinerant electrons [B]. Not only should the incorrect statement in the introduction be amended, but the correlated effects seen in these iridate materials are directly relevant to the subject of this paper and so should be discussed.

[A] Lefrançois, E. et al. Fragmentation in spin ice from magnetic charge injection. *Nat. Commun.* 8, 209 (2017).

[B] Pearce, M et al. Magnetic monopole density and antiferromagnetic domain control in spin-ice iridates. *Nat. Commun.* 13, 444 (2022).

2) The authors suggest that their observed transport anomaly is a charge response to the magnetic fluctuations in the transition region. They say that this picture "captures the positive sign of the anomaly". Please can the authors explain this comment more clearly?

This charge-response mechanism seems a reasonable possibility. However, there is another possibility that should be mentioned — a magnetocaloric effect. The large rate of change of resistance of the Bi₂Ir₂O₇ with temperature in the low-temperature region means that any small change in temperature caused by sweeping through the transition region could cause an anomaly in the transport which would likely have a very similar dependence on thickness and temperature as that observed. This is true even if the field is swept very slowly. The authors should discuss this.

3) On a related matter, the methods state that the field sweeps were performed with "a 0.2T/min sweeping rate; a wait time of more than 60 seconds and a step size of 0.05T". I am not completely sure what this means, please could the authors elaborate? The methods should also state the size and frequency of the current used in the lock-in transport measurements.

If the authors are able to respond effectively to these comments then I think this manuscript could be published in Nature Communications.

Reviewer #1 Comment # 1

This paper presents an experimental study on the magnetoresistance (MR) of the $\text{Bi}_2\text{Ir}_2\text{O}_7$ (BIO) / $\text{Dy}_2\text{Ti}_2\text{O}_7$ (DTO) interface. The DTO is a canonical spin-ice materials, while the BIO is a metallic magnet which also has the same pyrochlore structure. Unlike the spin-ice magnets $\text{Dy}_2\text{Ti}_2\text{O}_7$ or $\text{Ho}_2\text{Ti}_2\text{O}_7$, the itinerant electrons favor an all-in-all-out magnetic order in BIO. However, as demonstrated in this manuscript, the heterostructure formed from BIO and DTO provides a metallic thin layer whose electrical transport depend on the proximate spin-ice state in DTO, and hence can be controlled by the magnetic field.

In particular, the authors clearly showed an anomalous MR induced by a transition from the highly degenerate kagome ice phase to the fully polarized state in DTO. The experimental results and analysis seem solid to me. The reported MR anomaly is quite interesting, this work could potentially open a new avenue for meta-materials and quasi-2D systems based on spin-ice magnets.

I have a few questions/comments:

1. In Figure 2(b) and (d), the background MR (%) generally decreases with increasing magnetic field B at the temperature $T = 0.03\text{K}$. On the other hand, the MR (except for the anomaly at $B \sim 1.5\text{ T}$) in Figure 3(c) increases with increasing B -field in the temperature range $T = 0.45$ to 1K . Can the authors elaborate on why the trends of the background MR seem to be opposite at extremely low temperature ($T \sim 0.03\text{K}$) and close to the liquid-gas critical point. (Yet, interestingly, the anomaly persists in both regimes.)

Reply:

We thank the referee for reviewing the manuscript and recognize the value of our work. The overall measured MR of the thin BIO film is positive (increases with increasing magnetic field) at higher temperatures, such as that above 2K (see Fig. S2e). As the temperature becomes lower, a negative MR starts to emerge at lower magnetic field, and this negative region slowly expands in field with decreasing temperature. While the exact nature of this negative region is not yet clear, it may be related electron-electron interaction correction at 2D since it is more significant in the thinner films regardless of the substrate. Nevertheless, we do not find any strong correlation between the MR anomaly and the temperature/field range of the negative MR. In Figure 3(c), at $T = 0.45\text{K}$ to 1K , the negative region is still below 1T and therefore the MR anomaly $\sim 1.5\text{T}$ sits on

an MR background of positive slope. When further cooled to $T=0.03\text{K}$, the negative region expands enough to cover the measured field range, and the MR anomaly sits on a negative slope. If one measures the background MR to higher fields, like $>10\text{T}$, it will eventually turn positive. But that high field range is not relevant to the physics of the anomaly. We have added additional elaboration about this in the revised supplementary as: “As the temperature decreases... which is why the background MR in Fig. 2 appears to be negative. ... While the exact nature of this negative region is not yet clear, it may be related to the electron-electron interaction correction at 2D since it is more significant in the thinner films regardless of the substrate. Nevertheless, we do not find any strong correlation between the MR anomaly and the temperature/field range the negative region covers.”.

Reviewer #1 Comment # 2

2. Related to the above point is that: In the low-field kagome ice phase, the geometrical frustration results in a classical spin liquid due to the huge quasi-degenerate microscopic states governed by the kagome-ice rule. However, this is essentially a spin-disordered state. Generally, one would expect this additional disorder (either static or dynamical, due to fluctuations between different kagome ice configurations) would give rise to a larger resistivity, compared with that of the fully polarized state at high field?

Due to the rather distinct nature of the kagome-ice phase and the fully polarized state, one would expect the MR to be rather different in the two phases. However, as the authors noted, the background MR seems to follow a smooth curve (also different at low- T and T_c). Can the authors also elaborate on this point, namely the absence of two MR-behaviors in the low and high-field phases?

I guess one possibility is that the background MR comes almost entirely from the BIO layer, so it is insensitive to the spin-state on the DTO side or the interface. And the anomalous MR at $B \sim 1.5\text{ T}$ is an additional signal due to the interface?

Reply:

The referee has certainly raised a very interesting question about whether the carriers in BIO are sensitive to the different degrees of spin disorder of the DTO spin-states. What we would

like to point out is that our study is not to conclude that the carriers are insensitive to the spin state. The experimental fact, as the referee has noticed, is that the background MR does not seem to show significant difference in the low- and high-field phases. As the referee suggested, a possible reason is that the background MR from the BIO layer itself is much bigger than the difference induced by the spin states. In other words, the MR anomaly at 1.5 T is observable because it is significant enough to be resolved on the top of the background MR.

On the other hand, we could examine this question of whether spin disorder in DTO affects the BIO resistivity in a different way. That is by comparing the resistivity at the same field strength but applied along different directions. This approach exploits the fact that BIO shows isotropic MR in the absence of DTO influence and the DTO spin-state depends on the field direction. From the inset we added to Fig.2b in the revised manuscript, one can see that the resistivity with $B//[1-10]$ (i.e., 90 deg) at 5T is higher than that with $B//[111]$ (i.e., 0 deg). Since DTO remains a spin ice at 5T with $B//[1-10]$ but becomes a saturated ice with $B//[111]$, the larger resistivity in the former case is likely due to the spin disorder, which is consistent with the referee's speculation. We can also do the same comparison between $B//[110]$ and $B//[111]$ or $[11-1]$ at 5T from the data in Fig.2d. We plotted the same data in the new Fig. S2g without the vertical shift. One can see the resistivity is also larger with $B//[110]$ as a result of the increased spin disorder. Similarly, when comparing $B//[110]$ and $B//[001]$, the resistivity of the former is higher than the latter. This is consistent with the fact that the spin-ice degeneracy is completely lifted with $B//[001]$ even though the magnetic structure remains 2-in-2-out. This comparison is still not yet exactly a comparison with the Kagome ice state because the difference on a single MR curve is likely overshadowed by the MR background and the anomaly near H_c . But the overall trend is consistent with the expectation that the resistivity is higher with spin disorder. We have added additional elaboration about this in the revised manuscript.

Reviewer #1 Comment # 3

Finally, a minor comment on the terminology. The transition of DTO at $B \sim 1.5T$ is essentially a transition from 2-in-2-out kagome ice plateau to the fully polarized state. Indeed, this transition can be viewed as a condensation of monopoles (a statement which is less trivial at $T \sim T_c$). But otherwise, this is a very simple magnetic transition, which can be easily demonstrated

with one or two figures. I am not sure whether throwing terms like “monopole condensation” can help the general readers understand the underlying physics.

Reply:

We thank the referee for the suggestion. We have revised the statement “*To summarize, the experimental observation of an emergent MR anomaly in BIO that responds to and closely tracks the magnetic monopole condensation transition in insulating DTO...*” with “*To summarize, the experimental observation of an emergent MR anomaly in BIO that responds to and closely tracks the Kagome ice-to-saturated ice transition in insulating DTO*”.

Reviewer #2 Comment # 1

This paper presents results from an impressive materials fabrication achievement: growing a conducting pyrochlore epitaxially on top of a spin ice material. The authors demonstrate that the magnetic physics of the spin ice affects the transport properties of the conducting layer in that there is an anomaly in the magnetoresistance that corresponds to the metamagnetic transition in the spin ice. This is interesting and impressive as a materials growth advance, but I find the paper missing two critical points that would be required for publication:

1. The authors refer to a 3.4 nm and a 3.7 nm sample, but they never list all the samples that they measure or show data from the full list to allow for a sense of reproducibility in the results. This is critical, since the authors do demonstrate that the result disappears for large thicknesses as might have been expected, and some comparison would illuminate possible mechanisms. The SI should include such a list and side-by-side comparisons of the measured properties. This comparison should include some discussion of the quantitative aspects of the anomaly, as well as the qualitative similarities and differences.

Reply:

We thank the referee for reviewing the manuscript and appreciate the referee’s recognition of our work. Regarding the 1st question, we focused on the 3.4 nm sample in the manuscript since it was the one we completed the systematic angle-dependent measurements on. Such measurements in a dilution refrigerator took significant efforts and magnet time in NHMFL. It is impractical to repeat the complete set of measurements on many samples. To respond to the

referee's request, we have now included the MR results with B//[111] at 0.03K of another 3.9 nm BIO/DTO as well as a 4.0 nm BIO/DTO, in addition to the 3.2nm, 3.4nm, 23.8nm and 41.4 nm samples that we already shown. These are all the samples we were able to measure at temperatures close to 0.03K. The updated plot in Fig. S3 and Fig. S2f demonstrates the reproducibility of the MR anomaly. We have also included a list of synthesized heterostructures in Table S2 as the referee requested.

Reviewer #2 Comment # 2

2. The authors assert that the field from the spin ice cannot account for the MR anomaly because the sign is incorrect. I am not sure that is the case, given that the local fields may have interesting directional dependence, and I would have appreciated a bit more detailed explanation on this point. More importantly, however, the authors give no suggestions for what mechanisms could be coupling the two materials. Is there some correlated electron effect in the conductor that could be coming into play that somehow couples with the spin ice moments (hard to imagine)? Is the conduction happening through the spin ice in some electronic proximity effect (also hard to imagine)? Is there a strain effect associated with the transition in the spin ice (possible, but I would expect a step rather than a peak in the MR)? I can speculate as to what might be occurring here, but the authors need to include some discussion at a bare minimum for publication in any respectable journal in order to make a complete story. For publication in Nature Communications, I would expect that they would propose a mechanism and have some physics explanation for why it would produce a signal of the sort that they observe.

Reply:

We thank the referee for this suggestion. We would like to clarify that the field we referred to is the stray field from a magnet, which in this case is the DTO substrate as it has a significant magnetization when being magnetized by the applied field. We have updated the wording with “stray field” to avoid confusion. The local field the referee referred to as having directional dependence is an effective exchange field on the spin of the carrier from the local Ising moment, which has a quantum mechanical nature because it depends on the hybridization between the wave functions of the localized 4f-states and the itinerant 5d-state. As in many metals with local

moments, when the localized state is well below the Fermi level and the hybridization with the itinerant state is not strong, which is likely the case here, the coupling between the carrier and the localized electron can be effectively modelled as an exchange interaction of their spins. This mechanism is usually captured by the so-called Anderson model. In this model, correlation is only considered for the localized electrons/moments. This basically answers the referee's question about the coupling mechanism. The hybridization is probably what the referee referred to by "proximity effect". The difference in our structure from the usual metals with local moments is just that the hybridization only occurs at the interface. Since the spin ice is in the DTO single crystal substrate, the transition is the same as the bulk and there is no strain effect as demonstrated in our AC susceptibility data. We have added additional elaboration about this in the revised manuscript.

Reviewer #3 Comment # 1

This paper reports experiments on a heterostructure made up of two pyrochlores, the spin-ice insulator $Dy_2Ti_2O_7$, and the paramagnetic metal $Bi_2Ir_2O_7$. The authors make an excellent case that the first-order kagome-ice-to-saturated-ice transition in $Dy_2Ir_2O_7$ leads to an anomaly in the transport of the $Bi_2Ir_2O_7$. This observation highlights the very interesting possibility of devices in which the electrical detection of transitions between exotic spin states in frustrated insulators is possible. The paper can be published provided that the authors address the following comments.

1) In the introduction the authors state that "However, the spin-ice behaviors have been only observed in highly insulating compound, where the spins are hosted by highly localized electrons in the absence of itinerant carriers, such as the f -electrons of Ho^{3+} or Dy^{3+} ions in combination with the empty d shell of the Ti^{4+} valence state in the pyrochlore titanate. When introducing itinerant carriers, simply replacing the B-site with another element with a metallic configuration often ends up destroying the spin ice state. For instance, the correlated d -electrons in pyrochlore iridates not only exhibit an all-in-all-out ordering of their own at much higher temperatures, but also force the same magnetic order onto the A-site sublattice regardless of the rare earth element." This is not correct. For example, $Ho_2Ir_2O_7$ and $Dy_2Ir_2O_7$ both show spin-ice behaviour on the rare-earth sublattice at the same time as all-in-all-out order on the Ir sublattice. Indeed, interactions between the two sublattices have been shown to lead to interesting effects,

including (i) a fragmented monopole crystal ground state at low temperatures [A]; and (ii) at slightly higher temperatures, a response to changes in the monopole density that can be detected electrically via the itinerant electrons [B]. Not only should the incorrect statement in the introduction be amended, but the correlated effects seen in these iridate materials are directly relevant to the subject of this paper and so should be discussed.

[A] Lefrançois, E. et al. Fragmentation in spin ice from magnetic charge injection. *Nat. Commun.* 8, 209 (2017).

[B] Pearce, M et al. Magnetic monopole density and antiferromagnetic domain control in spin-ice iridates. *Nat. Commun.* 13, 444 (2022).

Reply:

We thank the referee for reviewing the manuscript and recognize the value of our work. The referee is correct that, while the Ir-ordering will force the rare earth sublattice to form an all-in-all-out ordering at relatively higher temperatures in rare earth pyrochlores, more complex behaviors may emerge when the temperature is low enough so that the rare earth-rare earth interactions become more important. We've now emphasized this point in the introduction as "When introducing itinerant carriers, simply replacing the B-site with another element with a metallic configuration often ends up interfering the spin ice state. ... In some cases, the interactions within the rare earth sublattice may modify the transport properties of the d -electron²²⁻²⁴ at lower temperatures, such as the fragmented monopole crystal ground state²⁵ as well as the monopole-density-related magneto-transportation²⁶. However, the spin state of the rare earth sublattice is often impaired by the energy scale of the d -electrons, which is much larger than the interaction."

We have added the relevant references as well. We hope by doing so would relieve the referee's concern.

Reviewer #3 Comment # 2

2) *The authors suggest that their observed transport anomaly is a charge response to the magnetic fluctuations in the transition region. They say that this picture "captures the positive sign of the anomaly". Please can the authors explain this comment more clearly?*

Reply:

This statement means that the picture we put forward explains why the MR anomaly is a positive peak instead of a negative dip or a step-like jump. More specifically, since the spins at the boundaries of the 2-in-2-out region and the saturated ice region are unstable and much easier to flip, they effectively act like scattering centers. The number of such scattering centers are expected to maximize at the critical field and decreases when away from the transition, leading to a peak-like change in resistivity. To make the relation between the unstable sites and the resistivity more explicit, we've added "The spins nearby the boundary are most unstable and fluctuating, thus potentially introducing extra scattering to the charge carriers when entering one region from another which increases the resistivity. Figure 3a shows a cartoon of how the density of these boundaries may increase and decrease across the critical regime, capturing the enhanced resistivity during the transition manifested by the positive sign and the cusp shape of the anomaly." in the text.

Reviewer #3 Comment # 3

This charge-response mechanism seems a reasonable possibility. However, there is another possibility that should be mentioned — a magnetocaloric effect. The large rate of change of resistance of the $\text{Bi}_2\text{Ir}_2\text{O}_7$ with temperature in the low-temperature region means that any small change in temperature caused by sweeping through the transition region could cause an anomaly in the transport which would likely have a very similar dependence on thickness and temperature as that observed. This is true even if the field is swept very slowly. The authors should discuss this.

3) On a related matter, the methods state that the field sweeps were performed with "a 0.2T/min sweeping rate; a wait time of more than 60 seconds and a step size of 0.05T". I am not completely sure what this means, please could the authors elaborate? The methods should also state the size and frequency of the current used in the lock-in transport measurements.

Reply:

The referee raised an interesting question regarding the potential magnetocaloric effect. By examining our data, this effect can be ruled out from several aspects. First, if the magnetocaloric effect is dominant, one would expect opposite effects on the resistance when up-sweeping and down-sweeping the field, which means the MR anomaly would have opposite signs. In Fig. R1 below, we demonstrate that this is not the case.

Figure R1 Magnetoresistance of the 3.2nm BIO/DTO heterostructure at 0.03K with field sweeping upward (black) and downward (red).

Secondly, in our work, we swept the magnetic field very slowly at 0.2T/min and incorporated 60s of wait time after the field reached each setpoint (i.e. 0.05T, 0.1T, 0.15T etc.) to let the sample reach thermal equilibrium. The results were thus collected after the actual sample temperature was stable with minimum fluctuation. Such an experimental procedure minimizes any magnetocaloric effect. This was discussed in more details in the supplementary.

In the revised manuscript, we've added "In addition, the observed anomaly could not be attributed to the magnetocaloric effect as it is expected to cause responses of opposite signs when sweeping the magnetic field upward and downward." in the discussion; we have also clarified the experimental details in the main text as "MR at 0.03 K was measured at the National High Magnetic Field Laboratory (NHMFL), using a 0.2T/min sweeping rate; the data were collected at field values of a 0.05T step size, after waiting for more than 60 seconds at each setpoint." and "The typical current applied was at the scale of 10 nA, with a frequency < 20 Hz." We hope by doing so would relieve the referee's concern.

List of major changes:

Main Text:

1. Page 3, paragraph 1. Replace “*When introducing itinerant carriers...This situation indeed applies to a broad class of frustrated magnets beyond spin ice.*” with “When introducing itinerant carriers, simply replacing the B-site with another element with a metallic configuration often ends up interfering the spin ice state...In some cases, the interactions within the rare earth sublattice may modify the transport properties of the d -electron²²⁻²⁴ at lower temperatures, such as the fragmented monopole crystal ground state²⁵ as well as the monopole-density-related magneto-transportation²⁶. However, the spin state of the rare earth sublattice is often impaired by the energy scale of the d -electrons, which is much larger than the interaction. This situation indeed applies to a broad class of frustrated magnets beyond spin ice.”.
2. Page 5, paragraph 1. Replace “*This anomaly was confirmed in multiple samples and has a reduced size in samples with a much thicker BIO film, indicating that the anomaly is caused by the interface with DTO (Figure S3a).*” with “This anomaly was confirmed in multiple samples (Table S2 and Figure S2f). It can also be seen that the anomaly has a reduced size in samples with a much thicker BIO film, indicating that the anomaly is caused by the interface with DTO (Table S2 and Figure S3a).”
3. Page 6, paragraph 1. Add “which is consistent with the fact that the anomaly is quickly shifted beyond the measured field range and the shape of the MR curve...”
4. Page 6, paragraph 1. Add “The angle-controlled transition is further illustrated in the inset of Figure 2b, where the MR at 5T is extracted and plotted against the field angle from [111] ($\theta=0^\circ$) to [1-10] ($\theta=90^\circ$). One can see a sharp increase of the resistivity around the critical angle θ_c ”

(~20°) due to the anomaly. Interestingly, the resistivity on both sides of θ_c shows a much smaller but visible difference, which can be attributed to the fact that DTO remains a spin ice at 5T with $\theta > \theta_c$ while becoming a saturated ice with $\theta < \theta_c$. The remnant spin disorder in the former is likely to elevate the resistivity compared with the latter.”

5. Page 7, paragraph 1. Add “Note that the observed positive cusp-like anomaly cannot be attributed to the DTO stray field on BIO ... and lead to a negative step-like MR jump instead.”
6. Page 7, paragraph 1. Add “In addition, the resistivity at 5T with $B//[110]$ is also larger than that of $B//[111]$ and $[11-1]$ (Figure S2g), consistent with the impact of the remnant spin disorder in the spin ice state as discussed with the previous configuration.”
7. Page 8, paragraph 1. Add “The spins nearby the boundary ... which increases the resistivity. Figure 3a shows a cartoon of how the density of these boundaries may increase and decrease across the critical regime, capturing the enhanced resistivity during the transition manifested by the positive sign and the cusp shape of the anomaly. As discussed earlier in the angle-dependent data, although the different degrees of spin disorder of the two states may still induce a difference in the resistivity outside the transition region, this effect is smaller than the anomaly likely due to the fact that local moments are more stable away from the transition.”
8. Page 8, paragraph 2. Replace “*similar resistance behaviors*” with “isotropic MR”; replace “*except*” with “without any”.
9. Page 9, paragraph 1. Add “Although a microscopic theory is yet to be developed to describe the interaction of the Dy moment and the BIO carriers, it likely depends on the hybridization between the localized f -states at the Dy sites and the itinerant states of Ir d -electrons since such an interaction is usually captured by the Anderson Model^{35,36} for metals with local moments.

The difference here is that the localized moments and the itinerant carriers are on opposite sides of the heterostructure and they couple across the interface.

10. Page 9, paragraph 1. Add “In addition, the observed anomaly could not be attributed to the magnetocaloric effect as it is expected to cause responses of opposite signs when sweeping the magnetic field upward and downward, which was not observed in our experiment.”
11. Page 10, paragraph 2. Replace “*magnetic monopole condensation*” with “Kagome ice-to-saturated ice transition”.
12. Page 13, paragraph 1. Add “the data were collected at field values of a 0.05T step size, after waiting for more than 60 seconds at each setpoint.” and “The typical current applied was at the scale of 10 nA, with a frequency < 20 Hz.”
13. Add four new references to the main text (ref. 25, 26, 35 and 36 in the revised manuscript).
14. Add an inset to Figure 2b.

Supplementary Material:

1. Add a ‘Sample Information’ section and the corresponding Table S2.
2. Page 2. Add “Similar negative MR at ultralow temperatures was observed in other samples as well, including the BIO/ Y₂Ti₂O₇ reference sample discussed below. While the exact nature of this negative region is not yet clear, it may be related to the electron-electron interaction correction at 2D since it is more significant in the thinner films regardless of the substrate. Nevertheless, we do not find any strong correlation between the MR anomaly and the temperature/field range the negative region covers.”
3. Add the MR with $\mathbf{B} // [111]$ at 0.03K for BIO/DTO samples with similar film thickness to Figure S2f to demonstrate reproductivity. Add new field configurations to Figure S2g.

REVIEWER COMMENTS

Reviewer #1 (Remarks to the Author):

The authors have satisfactorily addressed my comments/questions, as well as those from the other referees. They have included further experimental results on different field directions to elucidate the origin of the MR signal. I recommend the publication of this paper in Nature Communications.

Reviewer #2 (Remarks to the Author):

The authors have partially addressed my earlier concerns, but I would ask for further significant changes before publication.

1. The data in the SI comparing the samples should include the actual plots of the magnetoresistance (MR) data so that other workers can see the results and compare the results from different samples. Simply providing a summary table is a big step in the right direction, but insufficient. The authors have clearly measured the MR on all the samples, and they should share the data so that other workers can see the size and shape of the peaks for each of the samples. This should be a trivial addition to the SI, but an important one.

2. The theory explanation for the MR anomaly is appreciated, but they provide only minimal connection to the spin ice physics literature in that explanation. The physics of DyTiO and this transition in particular has been explored for the better part of two decades, and there is quite a bit understood about what happens in that system at the field-induced transition. For example, they connect the change in MR with an enhancement of fluctuations among the Dy moments that is reflected in the AC susceptibility peak. Such fluctuations have been studied in numerous AC susceptibility studies, some of which are cited, but also directly through noise measurement (see Dusad et al., Nature volume 571, pages 234–239 (2019) and related papers from that group, although that work is not conducted at the field-induced transition -- enhanced fluctuations at a field-induced transition are actually directly observed in artificial spin ice (Goryca et al. Phys. Rev. X 11, 011042 2021)). I am also surprised that they don't connect to (Nature Communications volume 11, Article number: 1341 (2020)), which directly models what might happen at the interface between an iridate pyrochlore and a titanate pyrochlore.

The authors should expand on the connections to the spin ice literature around spin fluctuations and the field-induced phase transition. I am not sure that the above papers are the best to cite, but some more connection is warranted for the broader readership of Nature Comm.

Reviewer #3 (Remarks to the Author):

I thank the authors for addressing my comments and those of the other referees. I am satisfied that they have answered my queries and recommend the paper now be published.

Reviewer #2 Comment # 1

The authors have partially addressed my earlier concerns, but I would ask for further significant changes before publication.

Reply:

We thank the referee for reviewing our manuscript and appreciate the referee's input.

1. The data in the SI comparing the samples should include the actual plots of the magnetoresistance (MR) data so that other workers can see the results and compare the results from different samples. Simply providing a summary table is a big step in the right direction, but insufficient. The authors have clearly measured the MR on all the samples, and they should share the data so that other workers can see the size and shape of the peaks for each of the samples. This should be a trivial addition to the SI, but an important one.

Reply:

We would like to point out that the plots of the magnetoresistance were already presented in the SI of the last submission as Figure S2f & S3a. This was explicitly indicated in the response to the corresponding comment in the last reply-to-referees. And the plots were referred to in the discussion in the "Sample information" section as "... Information about sample BD1 to BD6 can be found in Table S2. ... The MR data of samples with similar BIO thickness at 0.03K with B//[111] (Figure S2f) demonstrates excellent reproducibility of the MR anomaly. ... The size of the anomaly is decreased and eventually vanished when significantly increasing the BIO thickness (Figure S3a)."

Reviewer #2 Comment # 2

2. The theory explanation for the MR anomaly is appreciated, but they provide only minimal connection to the spin ice physics literature in that explanation. The physics of DyTiO and this transition in particular has been explored for the better part of two decades, and there is quite a bit understood about what happens in that system at the field-induced transition. For example, they connect the change in MR with an enhancement of fluctuations among the Dy moments that is reflected in the AC susceptibility peak. Such fluctuations have been studied in numerous AC susceptibility studies, some of which are cited, but also directly through noise measurement (see Dusad et al., Nature volume 571, pages234–239 (2019) and related papers from that group, although that work is not conducted at the field-induced transition -- enhanced fluctuations at a field-induced transition are actually directly observed in artificial spin ice (Goryca et al. Phys. Rev. X 11, 011042 2021)). I am also surprised that they don't connect to (Nature Communications volume 11, Article number: 1341 (2020)), which directly models what might happen at the interface between an iridate pyrochlore and a titanate pyrochlore.

The authors should expand on the connections to the spin ice literature around spin fluctuations and the field-induced phase transition. I am not sure that the above papers are the best to cite, but some more connection is warranted for the broader readership of Nature Comm.

Reply:

We agree with the referee that the spin ice physics in $\text{Dy}_2\text{Ti}_2\text{O}_7$ and related compounds has been studied for quite a while, which is exactly the reason $\text{Dy}_2\text{Ti}_2\text{O}_7$ single crystal is an excellent prototype for exploring electronic responses through the interface.

We thank the referee for providing the interesting references. These papers are mainly about monopole plasma and monopole gas, where the monopoles carry the thermal fluctuations of the system. It corresponds to the 2-in-2-out state of $\text{Dy}_2\text{Ti}_2\text{O}_7$ as shown in [Dusad et al., Nature volume 571, pages 234–239 (2019)] raised by the referee. The magnetoresistance anomaly observed in our study is however driven by the critical fluctuations of the transition to the saturated ice (3-in-1-out/1-in-3-out) state. Since this transition corresponds to monopole condensation, there are significant distinctions between its critical fluctuations and the thermal fluctuations far away from the transition. Similarly, this critical fluctuation of monopole condensation has a different nature from the monopole plasma that occurs between two 2-in-2-out states in [Goryca et al. Phys. Rev. X 11, 011042 2021], even though both transitions require magnetic field. The monopole gas at the iridate/titanate interface proposed by [Nature Communications volume 11, Article number: 1341 (2020)] requires the iridate to have the same rare earth site and the all-in-all-out order, which is not the case in our system. And the electronic conduction of iridate does not appear to be essential.

Nevertheless, the concept and method demonstrated in our work point to a new direction of electronically probing monopole plasma, monopole gas, and other phenomena of magnetic monopoles. To benefit the broad readership of Nature Communications as the referee suggested, we have now included discussions on the potential connection to the physics of these references in the main text as “Recently, the magnetic monopole dynamics was detected by a magnetization noise spectrometer in the spin ice state of DTO⁴²⁻⁴⁵ as well as during the transition between two 2-in-2-out states of artificial spin ice⁴⁶. A two-dimensional monopole gas was also proposed at the heteroepitaxial interface of the 2-in-2-out state and the all-in-all-out⁴⁷. Such behaviors may manifest in electronic transport, too, when interfaced with charge carriers, providing another detection method and a potential route to application.”

List of the changes:

1. Main text, Page 10, paragraph 2, add “Recently, the magnetic monopole dynamics was detected by a magnetization noise spectrometer in the spin ice state of DTO⁴²⁻⁴⁵ as well as during the transition between two 2-in-2-out states of artificial spin ice⁴⁶. A two-dimensional monopole gas was also proposed at the heteroepitaxial interface of the 2-in-2-out state and the all-in-all-out⁴⁷. Such behaviors may manifest in electronic transport, too, when interfaced with charge carriers, providing another detection method and a potential route to application.”
2. Supplementary Information, Page 1, replace “*Figure 3a*” with “Figure S3a”.